# Patterns of Public Spaces in Spanish Mediterranean Touristified Historic Centres Based on Their Activities: Case Study of Malaga

Francisco Conejo-Arrabal [1], Carlos Rosa-Jiménez [2,*] and Nuria Nebot-Gómez de Salazar [1]

1 Department of Art and Architecture, School of Architecture, University of Malaga, Plaza El Ejido 2, 29071 Malaga, Spain; fconejoarrabal@uma.es (F.C.-A.); nurianebot@uma.es (N.N.-G.d.S.)
2 Institute of Habitat, Territory and Digitalisation, Ada Byron Research Centre, University of Malaga, Calle Arquitecto Francisco Peñalosa 18, 29010 Malaga, Spain
* Correspondence: cjrosa@uma.es; Tel.: +34-951952661

**Abstract:** Historic centres are undergoing a series of urban transformations as a consequence of the processes of touristification, and they are mainly located in pedestrianised public spaces. The consequences of the touristification of public space are manifested in its privatisation via the occupation of catering locals and changes to the uses of adjacent buildings. Recent literature has studied the touristification of the neighbourhood unit in an exhaustive way, but it has only studied specific variables of the public space unit. Therefore, an exhaustive study is needed to bring these variables together regarding the public space unit. This study proposes a methodology for categorising public space in terms of use, with the aim of identifying different patterns of activities with respect to touristification. To this end, a system of use indicators is defined according to the public space and adjacent buildings. This methodology has been tested in the Historic Centre of Malaga, analysing a sample of 54 public spaces and categorising them into five different patterns. This categorisation could facilitate the planning and regulation by local administrations of activities in the public space of the Historic Centre.

**Keywords:** urban planning and politics; touristification; public space; cities recovery; indicator system; historic centre





## 1. Introduction

Historic centres are undergoing a series of urban transformations due to the high incidence of mass tourism [1–3]. Touristification is the driving force behind these changes in cities, which can be identified not only in the landscape [4], but also in the use of public space, in buildings, and in the way these spaces are inhabited [5,6]. These transformations threaten to erase the centrality by which they are characterised [7,8]. Their main consequences are the privatisation of public space, changes to commercial activity, population displacement, and the transformation of housing into tourist accommodation [9,10]. Many of these impacts have already been highlighted as common problems of tourist destinations [11] that need to be addressed.

The touristification of public space is an undesirable impact of the urban transformation processes of historic centres as a consequence of (1) the rehabilitation of historic buildings and (2) the pedestrianisation of public space. While the first has sought to consolidate the historic centre as a tourist attraction for its economic development [12], the second has sought to develop more sustainable mobility by reducing the use of private vehicles [13] and promoting the use of public space as a place of activity for citizens [14]. However, the lack of integration of social values in these projects [15] and the economic boom as a consequence of these processes [16] have led to various problems in cities' way of life [17]: on the one hand, commercial gentrification [18], which has led to changes in ground-floor activity resulting in a loss of traditional retail in favour of franchises [19], and on the other hand, a proliferation of accommodation and catering services. The main

consequence of this trend of change in activity towards catering has been the privatisation of public space [20] and the resulting problems of coexistence with the resident population.

In this situation, the hypothesis of the pedestrianised public space as a common place where citizens carry out daily, functional, expressive, and ritual activities that bring the community together is at risk. This socio-cultural dimension cannot develop if there is no control over the uses and dynamics that are occupying public space, directly affecting the residential condition. Carrión [21] concludes that residential use as well as public space use is a determining factor in integrating society. In these transformation processes, traditional uses must be maintained, residents must be protected, gentrification processes must be controlled, and cultural and economic diversity must be preserved.

In the literature, several studies have analysed touristification in public spaces with regard to specific variables such as the occupation of public areas [20] or ground-floor activity [22]. From an integral perspective based on different variables or uses, the studies consulted have been carried out in other territorial units such as the neighbourhood or district unit [23] or only from the perspective of the building. In these cases, the neighbourhood unit suffers from the problem of being too large an area for a detailed study, and it is not possible to perceive spatial relations of how the public space is inhabited or used. With respect to methodologies for categorising public space, the literature is limited. Specific studies have analysed the categorisation of ground-floor activity [24], residential squares [25], and squares in non-touristic historic centres [26], from which different typologies of patterns have been identified. In a study by Yoon and Park [24], three patterns were extracted according to type of ground-floor activity: commercial, mixed and specifically residential. In a study by Cueva-Ortiz [26], four patterns of activities that converge in public space were detected: symbolic, symbiotic, exchange-related, and civic. From all of them, it can be concluded that the uses of buildings (both in general and specifically on the ground floor) affect the habitability of public space as well as the activities that take place in it.

This study aims to determine patterns of public space in touristified historic centres based on their activities. The research is based on the hypothesis that the activities in plots that are adjacent to public space influence its habitability. It also assumes that urban spaces with similar activities share a common characterisation and problems with respect to touristification. Based on these aspects, a methodology has been defined with a theoretical and an empirical phase and applied to the case study of Malaga.

The first phase consisted of compiling indicators of touristification from the recent literature and redefining these indicators based on the territorial unit of public space. The results led to the classification of indicators in two parts: activities in the use of plots adjacent to the public space (analysed based on building and ground-floor activities) and activities in the public space (analysed based on the street/plaza and façades).

The second phase consisted of testing the indicators in the historic centre of Malaga. The maximum values of the indicators led to the analysis of a sample of 54 streets and squares and their categorisation into five patterns of public space: Representative, Commercial, Tertiary, Restoration, and Recreational. Each of the identified patterns has common characteristics with respect to touristification, ranging from commercial–franchised spaces (Representative) to the most touristified ones (Recreational). The identification of the patterns also led to the study of the relationship between them and the spatial logic with which they are articulated in the city. The resulting categorisation of public space can be a tool that helps local administrations to make decisions in the planning and regulation of activities that take place in historic centres in order to promote and develop their habitability in response to touristification.

## 2. State of the Art

### 2.1. Approach to the Issue from the Street Unit

The problem of touristification and its consequences for the habitability of public space has been studied by different authors from the perspective of the street unit, taking into

account adjacent buildings and public space. Rescalvo and Báez [10] define the different elements that identify this touristification in the urban landscape, but without quantifying them: conflict manifestations on façades (graffiti, speeches, banners), tourist attractions such as buildings or public space (points of interest, tourist accommodation, places where tourists congregate), changes in commercial use at ground-floor level (new shops, vacant ones, those maintaining their activity or being replaced), physical changes in public space (redevelopment, maintenance, restaurant terraces) and in buildings (façades, rehabilitation of private buildings, renovation of public buildings). Huerga-Contreras and Martínez-Fernández [25] show their methodology of analysis and interpretation of public space and establish quantitative indicators; first the physical characteristics of the public space and then the uses of adjacent buildings.

With the same objective, Cueva-Ortiz [26] develops a methodology for analysing squares in an historic centre from a physical, normative, and social point of view. The resulting categorisation shows four typologies: Symbolic (they are representative spaces with monuments and full of personal lived experiences), Symbiotic (tertiarised spaces with facilities), Exchange (predominantly commercial, both products and services), and Civic (spaces used by citizens as meeting and/or demonstration points). With a similar aim, Yoon and Park [24] show the categorisation of public space based on ground-floor activity, and the results show three distinct patterns: exclusively retail use, mixed-use (with retail and restaurant activity), and residential use.

## 2.2. Indicators for Touristification

With regard to the studies that quantify the problem of touristification by means of indicators, two main models have been found: studies of several variables (studying indicators referring to several variables) and specific studies of a single variable (studying a single use on the basis of several indicators).

### 2.2.1. Multi-Variables Studies

Table 1 lists the variables and indicators that study the uses of buildings and public space from the different units of study in the literature. It can be seen that Marín Cots's research [23] analyses different physical variables of tourism in Malaga (Spain) and responds to specific OMAU (Urban Environment Observatory in Malaga) reports based on ground-floor uses [22], the saturation of tourist accommodation [27], and the occupation of public space by terraces [28]. The relationship between ground-floor uses and the supply of tourist accommodation has been studied in several studies [29,30], which conclude that there is a proliferation of catering establishments linked to a greater supply of accommodation in very touristic places. In one case study of an area of Seville (Spain), the results show a transformation of ground-floor use by up to 39% [29].

### 2.2.2. Variable-Specific Studies

There are specific studies that analyse each variable in detail using indicators. Many of these indicators have their origin in the study of gentrification and its relationship with changes in the use of commercial activity, such as the study by Zukin et al. [31], which specifically analyses the trend of increasing franchising in two gentrified neighbourhoods in New York (NY, USA). Measuring franchising as a percentage is used as an indicator in different studies [22–24,32], while other studies use the number of franchised outlets relative to others [30]. Moreover, ground-floor activity is studied extensively in articles and reports [22,24]. The first shows the case of Seoul, an analysis of the entire ground floors of the studied streets to obtain indicators of the density and diversity of their premises versus residential density. The second shows the analysis carried out in Malaga to determine the activities, premises, and franchises in the Historic City area. Taking the methodology and data of OMAU [22] as a starting point, Santos-Izquierdo et al. [33] analyse the changes produced on the ground floor in the 2021 year (post-COVID-19 pandemic) using a $50 \times 50$ m grid for the same case study. The transformation of the commercial fabric is also

investigated in Lisbon (Portugal) [32] using the indicator % of number of premises, but this time in different territorial units: neighbourhood, street and commercial equipment. In general terms, the results show an increase in the activity of restaurants, commercial franchises and a decrease in local commercial activity.

**Table 1.** Indicators of variables of touristification in the consulted literature. Source: Authors.

| Variable | Indicator | Results | Territorial Unit | Name of Unit, City (Year) | Source |
|---|---|---|---|---|---|
| Residential use | % Residential use on ground floor | 43.47 | Public space | Seoulsup 2-gil St., Seoul (2014) | [24] |
| Activity on ground floor | % Commercial activity | 100 | Public space | Garosu-gil St., Seoul (2014) | [24] |
| | % Catering activity | 14.89 | Public space | Bangbae-ro 42-gil St., Seoul (2014) | [24] |
| | No. of local shops | 475 | Area | Historic City, Malaga (2017) | [30] |
| | No. of restaurants and accommodation | 306 | Area | Historic City, Malaga (2017) | [30] |
| | No. of restaurants | 252 | Neighbourhood | Cannaregio, Venice (2019) | [34] |
| | No. of shops | 882 | Neighbourhood | San Marcos, Venice (2019) | [34] |
| | No. of restaurants | 204 | Neighbourhood | Baixa and Chiado, Lisbon (2020) | [32] |
| | No. of shops | 261 | Neighbourhood | Baixa and Chiado, Lisbon (2020) | [32] |
| | % Premises that remain in activity | 61 | Area | Santa Catalina, Seville (2020) | [29] |
| | % Closed traditional premises | 21 | Area | Santa Catalina, Seville (2020) | [29] |
| | % Tourist premises | 18 | Area | Santa Catalina, Seville (2020) | [29] |
| | No. of premises/50 × 50 | 21 | Area | Historic Centre, Malaga (2021) | [33] |
| Franchises | % Franchises | 16 | Neighbourhood | Central Harlem, New York (2006) | [31] |
| | % Franchises | 72.82 | Public space | Garosu-gil St., Seoul (2014) | [24] |
| | No. of franchises | 53 | Area | Historic City, Malaga (2017) | [30] |
| | % Franchises | 100 | Block | Larios St., Malaga (2019) | [23] |
| | % Franchises | 40.6 | Neighbourhood | Chiado, Lisbon (2020) | [32] |
| Occupation of public space | % Occupation | 90 | Public space | Uncibay Sq., Malaga (2015) | [23] |
| | % Occupation | 19 | Public space | Alianza Sq., Seville (2019) | [20] |
| | % Occupation | 30 | Public space | Real Sq., Barcelona (2019) | [20] |
| | % Occupation | 1.85 | Public space | Batallas Sq., Valladolid (2022) | [25] |
| | Area of ccupation | 75400 | City | Barcelona (2022) | [35] |
| | No. of terraces | 7729 | City | Milan (2022) | [35] |
| Touristification of housing | % Airbnb apartments/ Total housing | 37.34 | Neighbourhood | Sol, Madrid (2018) | [36] |
| | % Tourist rental housing/ Total housing | 18.24 | District | Historic Centre, Seville (2019) | [37] |
| | % Tourist rental housing/Total housing | 6.51 | District | Historic Centre, Cadiz (2019) | [37] |
| | % Airbnb apartments/ Total housing | 47.55 | Neighbourhood | Alfama, Lisbon (2019) | [38] |
| Tourist accommodation | % Tourist accommodation | 53 | Block | Malaga (2018) | [23] |
| | No. of tourist rental housing | 20,837 | City | Madrid (2019) | [39] |
| | No. of tourist rental housing | 20,404 | City | Barcelona (2019) | [39] |
| | No. of tourist rental housing | 7233 | City | Valencia (2019) | [39] |
| | No. of tourist rental housing | 6284 | City | Seville (2019) | [39] |
| | No. of tourist rental housing | 6051 | City | Malaga (2019) | [39] |
| | No. of tourist rental housing | 1939 | City | Palm of Mallorca (2019) | [39] |
| | No. of tourist rental housing | 1151 | City | Bilbao (2019) | [39] |

**Table 1.** *Cont*.

| Variable | Indicator | Results | Territorial Unit | Name of Unit, City (Year) | Source |
|---|---|---|---|---|---|
| Tourist use | No. of point of interest | - | City | Seville | [40] |
| | No. of points of interest according to Tripadvisor | - | City | Madrid, Barcelona, Valencia, Seville, Malaga, Palm of Mallorca, Bilbao (2019) | [39] |

The occupation of public space indicator is also present in the literature, and is directly linked to ground-floor activity. Generally speaking, there are two forms of ephemeral occupation of public space: commercial occupation and occupation by restaurant terraces. However, the studies consulted focus on the problem of occupation by terraces due to their differentiation in the surface area of occupation. There is a consensus on how to measure this indicator, using the area of occupation and the public space itself as a territorial unit [4,20,23,25,28]. However, the study by O'connell et al. [35] analyses the evolution of terraces in the cities of Barcelona (Spain) and Milan (Italy) before and after the COVID-19 pandemic based on the surface area of terraces and the number of terraces, respectively. In a more detailed way, several studies [4,20,28] analyse compliance with the regulations in each case study via several indicators.

One of the problems of touristification that has intensified in recent years is the touristification of housing [9]. In recent literature, this has been measured via the percentage of the number of dwellings offered versus the total number [23,27,36–38,41]. However, in the vast majority of cases, this indicator is studied at the neighbourhood or district unit level, with the exception of the studies by OMAU [27], which do so at the block unit level. The data from these studies show exponential growth on the part of the indicator: Parralejo and Díaz-Parra [37] analyse Seville and Cadiz (Spain), which increased to 18.54%. Sequera and Nofre [38] in the Alfama district of Lisbon (Portugal), show a result of 47.55%, while Malaga (Spain) stands at 37% [41]. In view of this situation, only Cantabria and Extremadura (Spain) recognise the touristification of housing as an extra-hotel use. The containment measures implemented are based on zoning with use limitation: Barcelona (Spain) establishes a zoning of different levels for the entire accommodation supply, with a restriction in the Ciutat Vella area, and a maximum density of 1.48% no. of tourist rental housing/housing per block [42]. Madrid (Spain) establishes limitations on the total tourist accommodation supply (hotels, tourist apartments, and tourist rental housing) in three areas of 36%, and 13% in the last two areas of the total supply [43]. San Sebastian (Spain), also through zoning, does not allow its use in zones where non-residential uses do not exceed 30% (zone A), while it allows this use in zones B and C, with a maximum of 250 m$^2$ within the non-residential uses of each building [44].

## 3. Methodology and Case Study

### 3.1. Methodology

The aim of our research is to determine the categorisation of public space in touristified historic centres. A phased methodology is presented which analyses the problem from the territorial unit of public space. The methodology is structured in two phases: (1) a theoretical phase of definition of indicators based on a literature review and (2) an empirical phase where patterns are determined from a specific case study.

Phase 1. Definition of indicators and assessment. Based on the literature, indicators of touristification are redefined from the public space unit. These indicators are based on the hypothesis that the activity of the adjacent plots of land influences the uses of the public space, so we found two main groups: projection of the activity of the adjacent plots of land in the public space and activity in the public space.

Phase 2. Pattern determination. In the empirical phase, initially the selection of public spaces for the testing of indicators is carried out. For the selection, the starting conditions of these spaces are taken into account, such as pedestrianisation, the occupation of the public

space and the activity of public space. The delimitation of the study area within the historic centre is performed in two steps:

1.  Selection of the sample (identification of touristified public spaces). The pedestrianised public spaces of interest for the analysis are selected on the basis of two characteristics: high intensity of activity on the ground floor of the adjacent buildings and high occupation of the public space by terraces and exhibitors. The first is measured by the percentage of façade length of active premises (commercial and catering) per unit of public space. The second is measured by the percentage of occupation (area of restaurant terraces and commercial displays) per public space.
2.  Identification of the scope of each case of the exhibition. The scope of each sample case is identified, which is made up of the area of public space and the adjacent plots.

Subsequently, for the definition of the public spaces patterns, the maximum and minimum values of the proposed indicators are analysed, and the sample is categorised according to the results obtained.

### 3.2. Case Study of Historic Centre of Malaga

The following methodology has been tested in the Historic Centre (HC) of the city of Malaga (Spain). Malaga has a population of 579,076 inhabitants [45] and has 11 municipal districts, with District 1 Centre comprising the largest part of the HC. The HC is structured in three zones: a sector of Arrabales to the north, the central intramural Historic City and the Ensanche Heredia to the south, as shown in Figure 1.

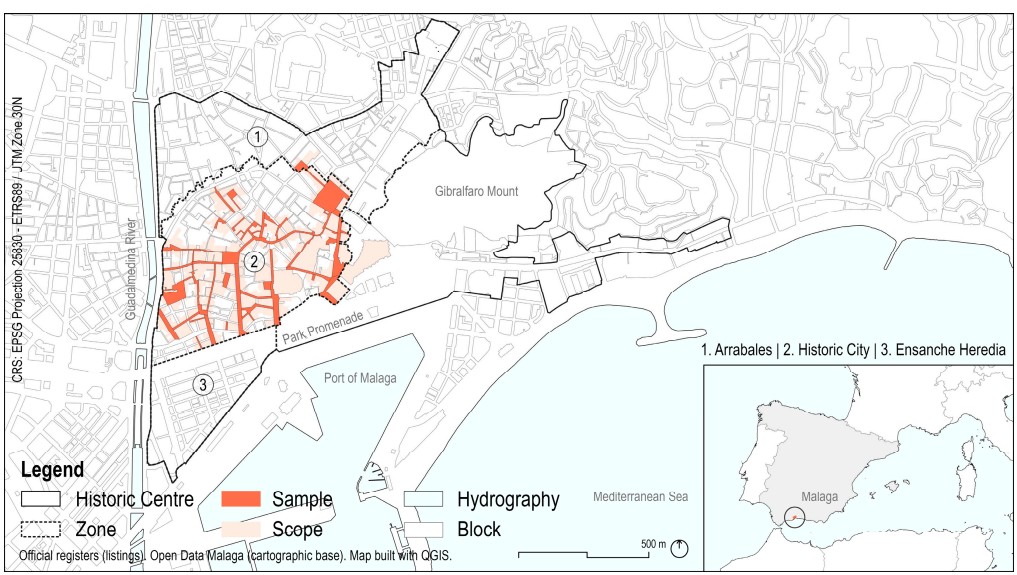

**Figure 1.** Case study of Historic Centre of Malaga. Source: Authors.

The processes of urban transformation have resulted in the almost-total pedestrianisation of the Historic City. These changes have taken place from 1990 to the current day during different periods [46], and although they initially presented the objective of revitalising public spaces and historic buildings and recovering the resident population [15], at present, they have caused "lack of accessibility, noise, lack of parking, misuse of public space, lack of neighbourhood facilities (. . .)" [15]. They have also led to a very significant population decline in the last decade, with a loss of up to 904 inhabitants [47].

Although the HC delimitation is broad, the tourist area corresponds to the Historic City, which has the highest concentrated density of urban regeneration projects. The problems of public space in this area are: (1) uncontrolled growth of tourist accommodation by platforms such as AirBnB, (2) the problems of intense pedestrianisation with a lack of planning (shortage of peripheral car parks or the privatisation of public space from restaurant terraces or commercial exhibitors), (3) the loss of population and the processes

of commercial gentrification, and (4) the impact on environmental sustainability (with the negative impact of noise pollution in areas where there is a concentration of nightlife and crowds for cultural activities that take place throughout the year).

## 4. Results

### *4.1. Determination of Indicators for the Touristification of Public Space*

The indicators are defined on the basis of the literature review, and they are grouped into two parts according to the object of study: Activities of the plots adjacent to the public space and Activities of the public space. In addition, in both parts, differentiated study elements are defined that affect the habitability of the public space, as shown in Figure 2.

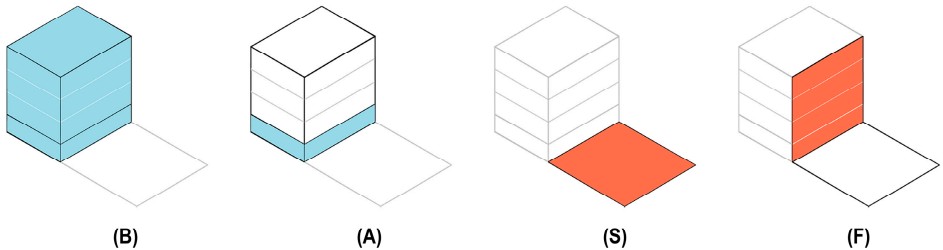

(B)　　　　　　　(A)　　　　　　　(S)　　　　　　　(F)

**Figure 2.** Elements study. Activities of the plots adjacent to the public space: (**B**) Building and (**A**) Activity on the ground floor. Activities of the public pace: (**S**) Street/Square and (**F**) Façade. Source: Authors.

#### 4.1.1. Indicators of Activities of the Plots Adjacent to the Public Space

The uses of the plots adjacent to each public space are studied based on two elements: general building activities (see Table 2: $B_{RE}$, $B_{ACC}$, $B_{APA}$ and $B_{TOU}$), and ground floor activity (economic activities take place mostly on the ground floor and directly affect the use of the public space and are therefore studied in detail, see Table 2: $A_{INT}$, $A_{TYP}$, $A_{COM}$, $A_{CAT}$ and $A_{FRA}$).

**Table 2.** Indicators of activities of the plots adjacent to the public space. Source: Authors.

| Element | Name | Equation | Source |
|---|---|---|---|
| Building | $B_{RES}$ \| Residential use<br>It measures the percentage of residential use area adjacent to the public space. | $\frac{Actual\ built-up\ area\ for\ residential\ use\ (m^2)}{Total\ built-up\ area\ (m^2)} \times 100$ | [48,49] |
| | $B_{ACC}$ \| Tourist accommodation<br>It measures the percentage of tourist accommodation area adjacent to the public space. | $\frac{Built-up\ area\ for\ H+TA+TRH\ (m^2)}{Total\ built-up\ area\ (m^2)} \times 100$ | [48,49] |
| | $B_{APA}$ \| Touristification of housing<br>It measures the percentage of TA and TRH area in relation to the residential area adjacent to the public space. | $\frac{Built-up\ area\ for\ TA+TRH\ (m^2)}{Total\ built-up\ for\ residential\ area\ (m^2)} \times 100$ | [48,49] |
| | $B_{TOU}$ \| Tourist use<br>It measures the percentage of tourist use area adjacent to the public space. | $\frac{Built-up\ area\ for\ MU+MO\ (m^2)}{Total\ built-up\ area\ (m^2)} \times 100$ | [48,50,51] |
| Activity | $A_{INT}$ \| Ground-floor activity intensity<br>It measures the percentage of active commercial and catering premises on ground floor adjacent to the public space. | $\frac{Built-up\ area\ for\ active\ premises\ (m^2)}{Total\ built-up\ area\ for\ ground\ floor\ (m^2)} \times 100$ | Fieldwork |
| | $A_{TYP}$ \| Type of activity<br>It measures predominant type of activity on the ground floor. | It is calculated from $A_{COM}$ and $A_{CAT}$. | |
| | $A_{COM}$ \| It measures the percentage of commercial premises in relation to the total premises. | $\frac{Built-up\ area\ for\ commercial\ premises\ (m^2)}{Built-up\ area\ for\ active\ premises\ (m^2)} \times 100$ | Fieldwork |
| | $A_{CAT}$ \| It measures the percentage of catering premises in relation to the total premises. | $\frac{Built-up\ area\ for\ catering\ premises\ (m^2)}{Built-up\ area\ for\ active\ premises\ (m^2)} \times 100$ | Fieldwork |
| | $A_{FRA}$ \| Franchises<br>It measures the percentage of franchises on the ground floor adjacent to the public space. | $\frac{Built-up\ area\ for\ franchises\ (m^2)}{Total\ built-up\ area\ for\ ground\ floor\ (m^2)} \times 100$ | Fieldwork |

1. $B_{RES}$ | Residential use. The objective of this indicator is the percentage of real residential use in each area. This takes into account the residential use obtained from the cadastre website [48], without taking into account the dwellings converted into tourist accommodation according to the official listings of Tourist Establishments and Services [49]. With respect to the literature, this indicator is used from the number of dwellings [36] or amount of population [34]. However, given the street unit and the possibility of a relationship with other uses, it is redefined on the basis of the built-up area.

2. $B_{ACC}$ | Tourist accommodation. The indicator of Marín Cots [23], is taken as a reference and adapted to the surface area. For this purpose, hotel tourist accommodation (H: hotel, hostel, guest house and guesthouse), tourist apartments (TA: buildings and complexes) and tourist rental housing (TRH: supply of complete dwellings and rooms) registered in official listings of Tourist Establishments and Services [49] are identified. The surface area of each property is then obtained according to the cadastre website [48].

3. $B_{APA}$ | Touristification of housing. The indicator of Parralejo and Díaz-Parra [37] is taken as a reference and adapted to surface area. To do so, tourist apartments (TA: buildings and complexes) and tourist rental housing (TRH: supply of complete dwellings and rooms) registered in official listings of Tourist Establishments and Services [49] are identified. The floor area of each property is then obtained according to the cadastre website [48].

4. $B_{TOU}$ | Tourist use. Based on the literature [40], this indicator is proposed to determine the tourist offer from the building in relation to the study unit. For this purpose, tourist buildings are taken to be those buildings that are accessible to the public or by means of tickets offered as museums (MU) [50] and monuments (MO) [51].

5. $A_{INT}$ | Ground-floor activity intensity. This indicator aims to measure the number of existing premises in each area. For this purpose, it takes other studies as a reference, but the indicator is redefined based on the surface area of existing premises compared to the total ground floor [22,29,32]. For testing in 2022, the methodology of OMAU [22] is used, and updated through fieldwork.

6. $A_{TYP}$ | Type of activity on the ground floor. It aims to define the specificity of the ground-floor activity with respect to the total built surface of the active premises. Although previous studies take into account the typology of premises [22,29,32], **$A_{TYP}$** allows the activity of each public space to be categorised into commercial, catering, or mixed according to $A_{COM}$ and $A_{CAT}$. For testing in 2022, the methodology of OMAU [22] is used, and updated through fieldwork.

7. $A_{FRA}$ | Franchises. The indicator of Marín Cots [23] is taken as a reference, and the scope is adapted to the one that refers to the street unit. For testing in the year 2022, the methodology of OMAU [22] is used, and updated through fieldwork.

### 4.1.2. Indicators of Activities of the Public Space

The activities of the public space are studied via two elements: the public space through its surface area (see Table 3: $S_{OCC}$) and the total façade of the public space through its length ($F_{TOU}$):

**Table 3.** Indicators of activities of the public space. Source: Authors.

| Element | Name | Equation | Source |
|---|---|---|---|
| Street/Square | $S_{OCC}$ \| Occupation of public space<br>It measures the percentage of occupation by exhibitors and terraces on the public space. | $\frac{Area\ of\ Terraces + Exhibitors\ (m^2)}{Area\ of\ public\ space\ (m^2)} \times 100$ | Fieldwork |
| Façade | $F_{TOU}$ \| Tourist façade<br>It measures the percentage of length of tourist façade in relation to the total length of public space. | $\frac{Length\ of\ tourist\ façade\ (m)}{Total\ length\ of\ public\ space\ (m)} \times 100$ | [48,50,51] |

1.  $S_{OCC}$ | Occupation of public space. The following indicator is taken as a reference of Elorrieta Sanz et al. [20], although the area occupied by commercial exhibitors is added, as already done by Huerga-Contreras and Martínez-Fernández [25], in addition to the restaurant terraces. The data for the study year are obtained through fieldwork.
2.  $F_{TOU}$ | Tourist façade. This measures the percentage of the length of the tourist façade with respect to the total length of each area. This indicator provides the quality of the public space as a focus of tourist attraction. Recent literature already takes into account that rehabilitated areas are likely to generate tourist attractions [10]. However, this indicator, beyond the rehabilitated surface area, quantifies it on the basis of the façades of museums [50], monuments [51] and advertised public spaces.

### 4.2. Patterns of Public Spaces

The indicators have been tested in the sample of the Historic City of Malaga in 2022 (data closed on 31 December). The indicators were evaluated quantitatively by ranges according to the maximum values observed in the literature. The $A_{TYP}$ indicator has been assessed qualitatively on the basis of the data obtained from $A_{COM}$ and $A_{CAT}$. The results of the indicators, as shown in Figure 3, reveal a polarisation of two different areas (commercial and catering) in the city and mixed spaces, which are situated as connecting spaces between the two areas. On the one hand, the area with a predominance of commercial activity has generally high values for franchises ($A_{FRA}$). Moreover, high values for tourist accommodation, and specifically for the touristification of housing ($B_{APA}$). Spaces with a predominance of catering activity have high values for occupation of public space ($S_{OCC}$) and tourist accommodation ($B_{ACC}$). However, the value of franchising is low compared to the previous one. The public spaces connecting these areas present a mixed activity, and their indicators are influenced by the adjacent spaces they connect.

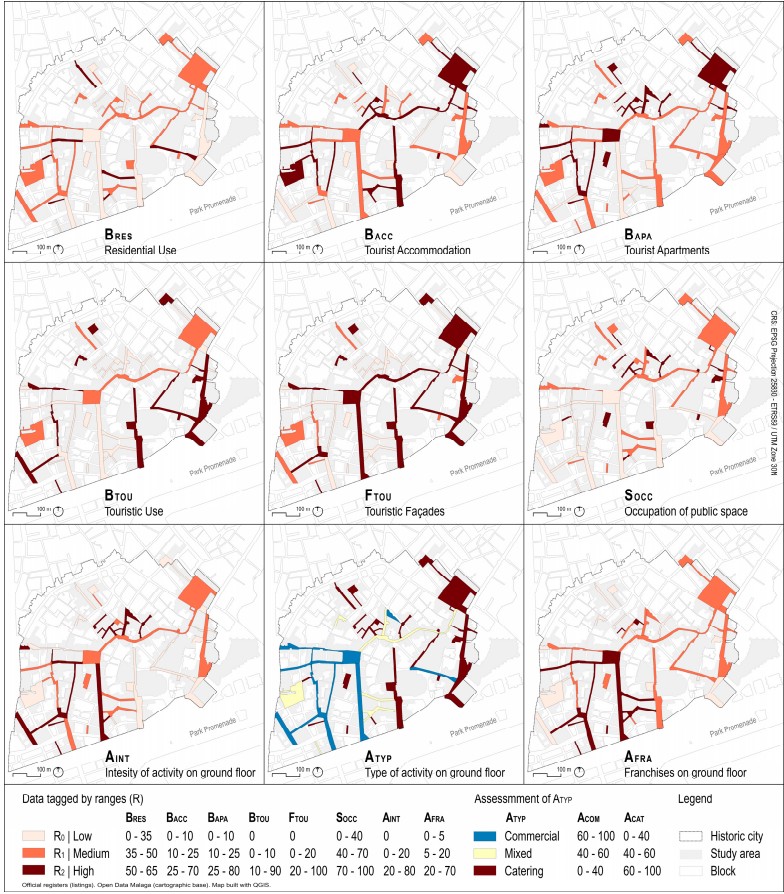

**Figure 3.** Testing and assessment of the indicators in the Malaga Historic City. Source: Authors.

The similar values of the sample indicators show a categorisation of five spatial patterns according to activities: two specifically commercial (Representative and Commercial), two catering (Catering and Recreational) and one tertiary (Tertiary). This categorisation is shown in Figure 4, as well as the different public spaces and areas belonging to each one.

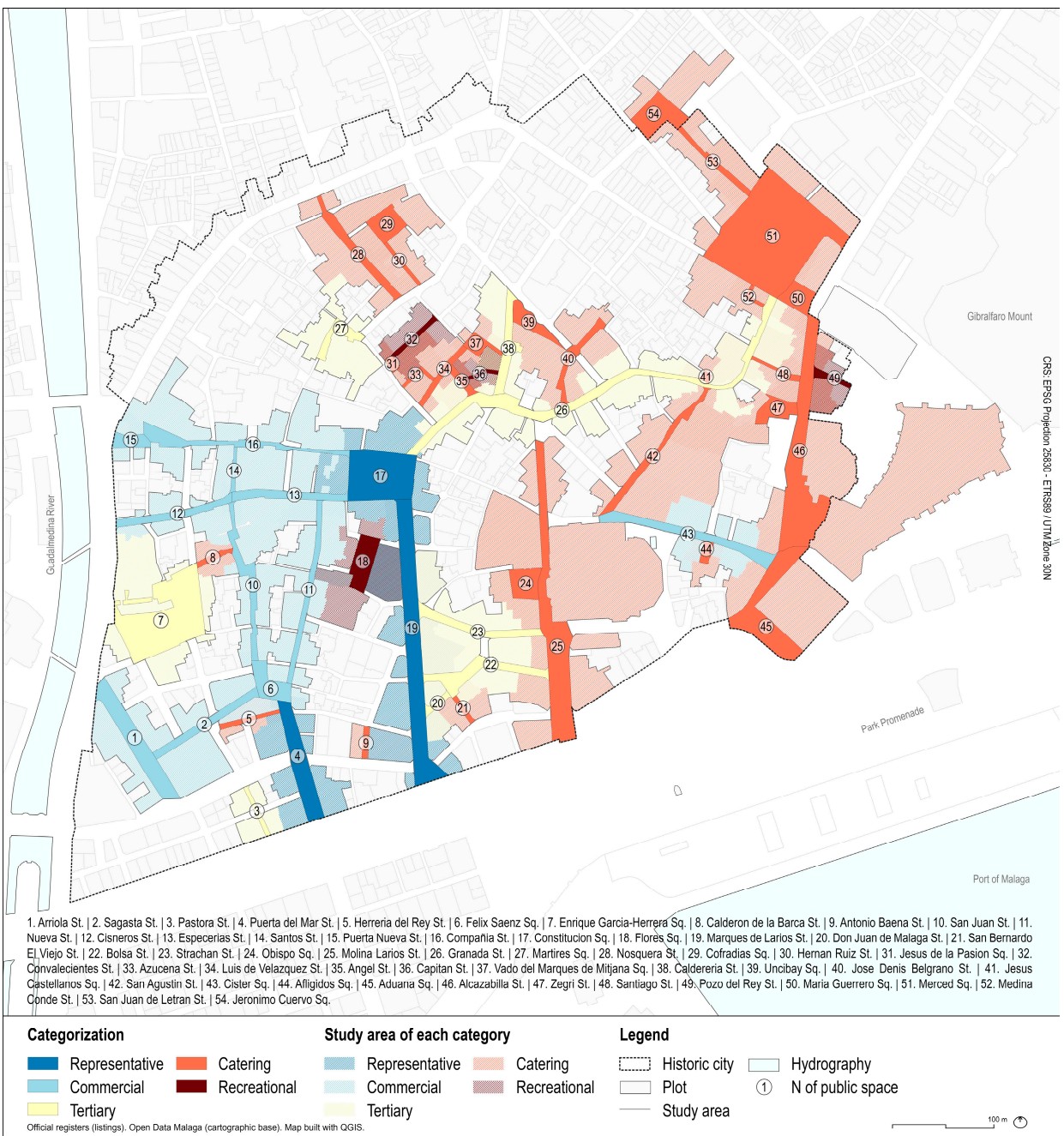

1. Arriola St. | 2. Sagasta St. | 3. Pastora St. | 4. Puerta del Mar St. | 5. Herreria del Rey St. | 6. Felix Saenz Sq. | 7. Enrique Garcia-Herrera Sq. | 8. Calderon de la Barca St. | 9. Antonio Baena St. | 10. San Juan St. | 11. Nueva St. | 12. Cisneros St. | 13. Especerias St. | 14. Santos St. | 15. Puerta Nueva St. | 16. Compañia St. | 17. Constitucion Sq. | 18. Flores Sq. | 19. Marques de Larios St. | 20. Don Juan de Malaga St. | 21. San Bernardo El Viejo St. | 22. Bolsa St. | 23. Strachan St. | 24. Obispo Sq. | 25. Molina Larios St. | 26. Granada St. | 27. Martires Sq. | 28. Nosquera St. | 29. Cofradias Sq. | 30. Hernan Ruiz St. | 31. Jesus de la Pasion Sq. | 32. Convalecientes St. | 33. Azucena St. | 34. Luis de Velazquez St. | 35. Angel St. | 36. Capitan St. | 37. Vado del Marques de Mitjana Sq. | 38. Caldereria St. | 39. Uncibay Sq. | 40. Jose Denis Belgrano St. | 41. Jesus Castellanos Sq. | 42. San Agustin St. | 43. Cister Sq. | 44. Afligidos Sq. | 45. Aduana Sq. | 46. Alcazabilla St. | 47. Zegri St. | 48. Santiago St. | 49. Pozo del Rey St. | 50. Maria Guerrero Sq. | 51. Merced Sq. | 52. Medina Conde St. | 53. San Juan de Letran St. | 54. Jeronimo Cuervo Sq.

**Categorization**
- Representative
- Commercial
- Tertiary
- Catering
- Recreational

**Study area of each category**
- Representative
- Commercial
- Tertiary
- Catering
- Recreational

**Legend**
- Historic city
- Plot
- Study area
- Hydrography
- ① N of public space

Official registers (listings). Open Data Malaga (cartographic base). Map built with QGIS.

100 m

**Figure 4.** Patterns of public space in the Historic City of Malaga. Source: Authors.

1. Representative. This pattern corresponds to spaces where franchised commercial activity predominates over other activities. This characteristic can be seen in the façades, which mostly have shop windows integrated into the façade of the building, and do not occupy the public space with exhibitors. This pattern shows a scarcity of residential use due to the intense tertiarisation in offices they suffer from. In addition, they are tourist areas of the city, so it is common to find representative street furniture such as sculptures, fountains, etc. in this type of space.

2. Commercial. This pattern corresponds to spaces with intense commercial activity. In contrast to pattern 1, this pattern occupies public space with exhibitors, as well as the use of other systems such as showcases and shop windows added to façades to advertise their activity. According to the sample, this pattern presents a higher percentage of surface area destined for residential use. However, some specific problems are identified: the proliferation of franchises, and the touristification of housing through tourist apartments.

3. Tertiary. This pattern corresponds to mixed-activity spaces, i.e., commercial and catering. Both activities occupy the public space with displays and terraces, respectively. Therefore, the public space is characterised by being very busy due to its activity. Although these are areas with a higher percentage of residential use than other patterns, the high intensity of activity can generate problems of coexistence with the residents of these spaces. In some cases, two problems are identified: the proliferation of franchises in commercial activity and high values in the touristification of housing.

4. Catering. This pattern corresponds to areas of catering activity. This has a direct impact on residential use, which presents the problem of a growing supply of tourist accommodation and, specifically, the touristification of housing. In addition, public space in most cases presents high occupancy values due to terraces and the different elements of their furniture: tables, chairs, or awnings, among others.

5. Recreational. This pattern corresponds to areas suffering from a saturation of tourist accommodation and restaurant activity. Residential use is minimal due to the touristification of housing. Public spaces are crowded by a proliferation of terraces, and therefore, mobility conflicts are common due to the narrowness of the passageways between them. Moreover, their location tends to be in smaller public spaces (section or surface area) close to others with tourist attractions.

## 5. Discussion

The sample analysed shows data on the state of each public space with respect to each pattern. On the one hand, the spaces of Pattern 1 present similar characteristics with respect to the literature regarding symbolic spaces [26]. It also shows the consequences of high tertiarisation in parallel with a reduction of residential use. This condition is similar to the process that the centre of Valencia (Spain) is undergoing [52]. With respect to ground-floor activity, it is noted that the indicators show a specificity of commercial franchises, as result of this process of tertiarisation, as well as gentrification. This fact coincides with other authors' results [24,31]. Actually, the similarity of the franchise values in Garosu-gil St. (Seoul) [24] is quite significant, with a value of 72%, very close to that of Larios St. in Malaga, with 65.87%, as shown in Table 4. However, Marín Cots [23] defines 20% franchising as a high value and, in this pattern of spaces, the values of this indicator are much higher.

Pattern 2 is mainly composed of commercial use activity and higher residential use, which corresponds to the interchange spaces [26]. It also has high values of some indicators such as $B_{ACC}$, $B_{APA,}$ and $A_{AFRA}$, although the most representative is the capacity to accommodate residential use, as is the case of Especerias St. at 50.32% according to Table 4. This data may be of interest in order to establish measures to protect residential use. Local regulations, such as those of San Sebastian (Spain) [44], do not allow for the use of tourist accommodation in areas with a predominant residential use of 70%, which is similar to the example in the sample. In fact, Especerias St. has a housing touristification of 19.61%, which suggests that it could have previously shown similar percentages in the residential aspect. The high value in $A_{ACC}$ and $A_{AFRA}$, at 10% and 20% and desirable in the thesis of Marín Cots [23], shows the trend that these spaces are facing with respect to touristification and gentrification, respectively.

**Table 4.** Results of indicators in public spaces of each pattern. $B_{RES}$: Residential use, $B_{ACC}$: Tourist accommodation, $B_{APA}$: Touristification of housing, $B_{TOU}$: Touristit Use, $F_{TOU}$: Tourist Façade, $S_{OCC}$: Occupation of public space, $A_{INT}$: Intensity of activity on ground floor, $A_{TYP}$: Type of activity, and $A_{FRA}$: Franchises. Source: Authors.

| Indicators | 1. Representative Marques de Larios St. | 2. Commercial Especerias St. | 3. Tertiary Granada St. | 4. Catering Hernan Ruiz St. | 5. Recreational Capitan St. |
|---|---|---|---|---|---|
| $B_{RES}$ | 26.82 | 50.32 | 43.86 | 38.11 | 18.90 |
| $B_{ACC}$ | 11 | 14.51 | 25.20 | 25.04 | 50.39 |
| $B_{APA}$ | 3.42 | 19.61 | 24.94 | 39.66 | 72.72 |
| $B_{TOU}$ | 0 | 0 | 2.17 | 0 | 0 |
| $F_{TOU}$ | 100 | 0 | 23.34 | 0 | 0 |
| $S_{OCC}$ | 0.90 | 0.18 | 17.79 | 25.14 | 29.05 |
| $A_{INT}$ | 0.90 | 72.35 | 65.47 | 55.57 | 91.53 |
| $A_{TYP}$ | COM | COM | MIX | CAT | CAT |
| $A_{FRA}$ | 65.87 | 37.77 | 17.35 | 0 | 0 |

Pattern 3 is at an intermediate level of indicators. There is a diversity of activities on the ground floor and high values for residential use, corresponding to the symbiotic spaces [26]. Moreover, this diversity of ground-floor activity corresponds to the values obtained in cluster 1 in Seoul [24]. The value of intensity of use on the ground floor of Granada St. is 65.47%, very similar to the 56.53% of Seoulsup 2-gil St. in Seoul, as well as heterogeneity in terms of activity. Even so, the case of Pattern 3 shows average values in indicators, to which attention should be paid because of their tendency to be specified to patterns 2 or 4. Furthermore, in both cases, there is a generalised decrease in residential use, and an increase in tourist accommodation, closely linked to ground-floor activity, as shown by the results of specific research [29,34,53].

With less diversity in terms of ground-floor activity, Pattern 4 refers to public spaces with a higher percentage of restaurant premises. The case analysed in Table 4, Hernan Ruiz St., shows high values for the occupation of public space and tourist accommodation and medium values with respect to residential use. It may correspond to the literature with so-called exchange spaces [26], as in Pattern 2. However, this combined trend of increasing values of $B_{ACC}$, $B_{APA}$ and the catering activity of ground floor are consequences of the process of touristification in which they are being immersed and which can be observed via low residential use. In addition, a very high value of $B_{APA}$ with 39.66% together with $S_{OCC}$ with 25.14% is identified in comparison to this study [23], which sets maximums of 10% and 20%, respectively.

Pattern 5 shows the maximum values of the touristified spaces. The case of Capitan St., as can be seen in Table 4, shows a very significant deterioration in residential use, mainly due to the process of residential tourism. In addition, it is accompanied by high values in $A_{INT}$ and $S_{OCC}$ and the specificity of restaurant activity. Specifically, $S_{OCC}$ presents a very similar occupancy rate to that of Barcelona (Spain) [20], with 30%. The high values in these indicators respond to a radicalisation of Pattern 4, studied in other cities [54] and perfectly identifiable in the literature as spaces of conflict [11].

In addition, the location of these patterns in the urban fabric of Malaga shows how spaces of different categories are connected. Firstly, Pattern 1 is identified in the most representative spaces of the city, and adjacent to these are spaces of Pattern 2, as shown in Table 5. In many of them, high values are observed in indicator $A_{AFRA}$, showing a possible tendency towards Pattern 1. In contrast, Pattern 5 is located in smaller spaces surrounded by spaces of Pattern 4. The influence of crowded areas of catering and tourist accommodation require examples of how the areas of Pattern 4 can evolve if no limiting measures are taken. On the other hand, Pattern 3 is located in those areas that unite highly

differentiated areas in terms of patterns such as 1 or 2 and 4. With values closer to those desirable in the literature, there are also places with a considerable residential use suitable for their conservation and development.

**Table 5.** Examples of public spaces of each pattern. $B_{RES}$: Residential use, $B_{ACC}$: Tourist accommodation, $B_{APA}$: Touristification of housing, $B_{TOU}$: Touristic Use, $F_{TOU}$: Tourist Façade, $S_{OCC}$: Occupation of public space, $A_{INT}$: Intensity of activity on ground floor, $A_{TYP}$: Type of activity, and $A_{FRA}$: Franchises. Source: Authors.

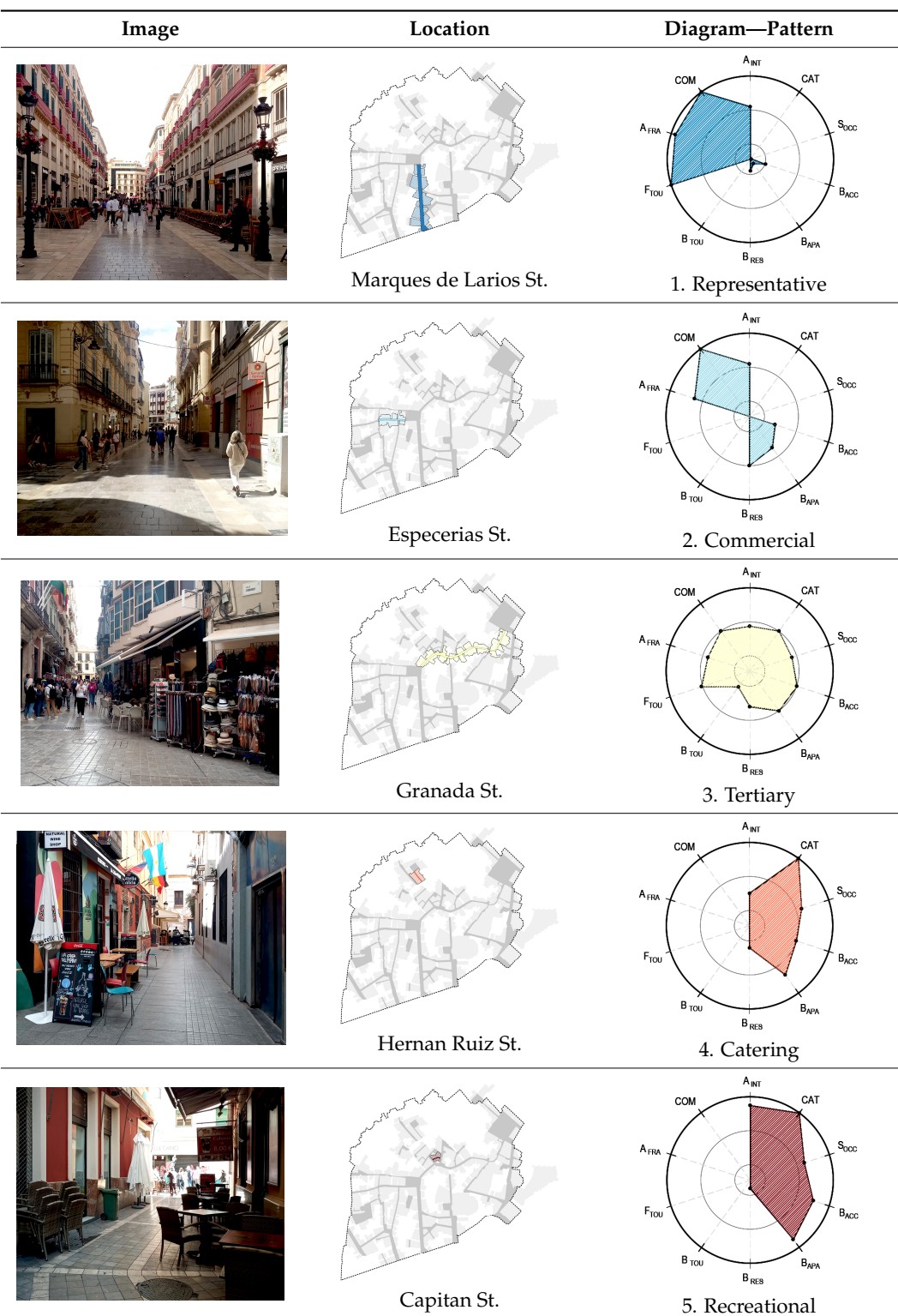

| Image | Location | Diagram—Pattern |
|---|---|---|
| | Marques de Larios St. | 1. Representative |
| | Especerias St. | 2. Commercial |
| | Granada St. | 3. Tertiary |
| | Hernan Ruiz St. | 4. Catering |
| | Capitan St. | 5. Recreational |

## 6. Conclusions

The study provides a methodology for categorising public space by means of the physical variables of touristification. In addition, indicators have been compiled and redefined for extrapolation to the territorial units of public space and from the study of activities. The study of these variables in an integral way has allowed us to take into account the uses of the buildings, ground-floor activities, façade, and the public space itself.

The sample analysed shows the existing relationship between the specificity of the hotel industry and public space and its repercussions on the decrease in residential use. It also shows the transformation of commercial activity, caused by the rise of franchises or the increase in catering activity. At the methodological level, the categorisation into five patterns has allowed for the diagnosis of these processes of touristification in the different public spaces analysed. Therefore, the study from the territorial unit of public space has made it possible to diagnose the degree of touristification according to activities, and it could be useful for an administration applying specific policies and measures for each pattern.

The research presents several limitations that have been detected in its course, and which could be developed in future studies: (1) Firstly, the tertiarisation by offices could be added as an indicator, to have information about economic activities that are not located on ground floor. (2) A comparative analysis of each pattern with respect to the physical characteristics of the public space. This relationship can conclude whether normative design parameters such as size, furniture and materiality influence the activities that are taking place in it. (3) Finally, this comprehensive study of activities provides quantitative information on public space, but it should be complemented with qualitative information on how citizens experience/use each pattern.

**Author Contributions:** Conceptualization, F.C.-A., N.N.-G.d.S. and C.R.-J.; data curation, F.C.-A.; formal analysis, F.C.-A. and N.N.-G.d.S.; funding acquisition, C.R.-J.; investigation, F.C.-A.; methodology, F.C.-A. and C.R.-J.; resources, F.C.-A.; supervision, N.N.-G.d.S. and C.R.-J.; validation, F.C.-A. and N.N.-G.d.S.; visualization, F.C.-A.; writing—original draft preparation, F.C.-A.; writing—review, and editing, N.N.-G.d.S. and C.R.-J. All authors have read and agreed to the published version of the manuscript.

**Funding:** This article was funded by FEDER funds (UMA20-FEDERJA-131), Andalusian Govern for Development, Infrastructures and Spatial Organization (UMA 20.01). The first author is supported by the Spanish Ministry of Universities through a PhD grant (FPU21/04662). The open access fee is funded by University of Malaga/CBUA.

**Data Availability Statement:** Data is contained within the article.

**Acknowledgments:** The authors would like to thank Lola Dumesnil for his help in the fieldwork and Francisco José Chamizo-Nieto for his help in the definition of indicator system. The authors are grateful to anonymous reviewers for their helpful comments and suggestions, which have made an important contribution to the final form of this article.

**Conflicts of Interest:** The authors declare no conflict of interest.

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
