# Peer review of "Patterns of Public Spaces in Spanish Mediterranean Touristified Historic Centres Based on Their Activities: Case Study of Malaga"

_land, doi:10.3390/land12081546_

Round 1

Reviewer 1 Report

The document is interesting and valuable for those in charge and interested in the analysis and reconfiguration of Historic Centers with similarities to the case of Malaga.

From my point of view, I consider that the title does not correspond to what the document shows since it only emphasizes a single case study; in my opinion, it should be reconsidered since it seems that the indicated patterns will serve different historical centers, which is not correct since the results are limited.

On the other hand, a high percentage of the references used are focused on Spain, it is understood that the case study is from this country, but the title emphasizes "touristified historic centers" as if the results were based on more than a case study.

The authors should review and reconsider how they reference other works since it is not homogeneous throughout the document, so it is recommended that they use the same typology for all cases. i.e., in some, they emphasize the authors' names or places of the study in question; in others, they only indicate the reference. It is also recommended to review the writing of the entire text in order to have more precise readability.

It is unclear the period to which the results correspond. Likewise, it would be convenient to add an image in the methodology in which the study area is indicated (it would help to have better document readability).

Table 4 should be improved since readability is lost; making more than one figure of this would be convenient.

Despite the preceding, the manuscript is significant and may generate interest in different areas. Therefore it should continue in the review process for its possible publication in this journal.

Reviewer 2 Report

Dear authors,

This is an interesting research on a topic that presents many challenges to urban management and public interest. The paper is well organized, the revision of literature is update, the methodology is appropriate, and the results are useful. 

Author Response

Authors would like to express their gratitude to Reviewer #2 for your revision and your comment

Reviewer 3 Report

The paper is interesting and well described. The methodology is clear and well explained.

No further comments.

Author Response

Authors would like to express their gratitude to Reviewer #3 for your revision and your comment